# A large-scale high-resolution numerical model for sea-ice fragmentation dynamics

Jan Åström[1], Fredrik Robertsen[1], Jari Haapala[2], Arttu Polojärvi[3], Rivo Uiboupin[4], and Ilja Maljutenko[4]

[1]CSC – It center for science Ltd. P.O. Box 405 FI-02101 Espoo, Finland
[2]Finnish Meteorologìcal Institute, Helsinki, Finland
[3]Aalto University, School of Engineering, Department of Mechanical Engineering, P.O. Box 14100, FI-00076 Aalto, Finland
[4]Tallinn University of Technology, Department of Marine Systems, Akadeemia tee 15a, Tallinn 12618, Estonia

**Correspondence:** astrom@csc.fi

**Abstract.** Sea ice motion and fragmentation forecasts are of vital importance for all human interaction with sea ice, ranging from indigenous hunters to shipping in polar regions. Sea ice models are also important for simulating long term changes in a warming climate. Here we apply a discrete element model (HiDEM), originally developed for glacier calving, to sea ice break-up and dynamics. The code is highly optimized to utilize high-end supercomputers to achieve extreme time and space resolution. Simulated fracture patterns and ice motion are compared to satellite images in the Kvarken region of the Baltic Sea in March 2018. A second application is ice ridge formation in the Gulf of Riga. With a few tens of graphics processing units (GPUs) the code is capable of reproducing observed ice patterns, that in nature may take a few days to form, over an area $\sim 100km \times 100km$, with an $8m$ resolution, in computations lasting $\sim 10$ hours. The simulations largely reproduce observed fracture patterns, ice motion, fast ice regions, floe size distributions, and ridge patterns. The similarities and differences between observed and computed ice dynamics and their relation to initial conditions, boundary conditions and applied driving forces are discussed in detail. The results reported here indicate that HiDEM has the potential to be developed into a high-resolution detailed model for sea ice dynamics over short time scales, which combined with large-scale and long-term continuum models may form an efficient framework for sea ice dynamics forecasts.

## 1   Introduction

Reliable forecast models for ice dynamics are of vital importance for all human activities related to sea ice. Indigenous hunters in the Arctic can move fast over long distances across level landfast ice, while travelling on drift ice or on land can be immensely more difficult. Similarly, sustainable and safe winter navigation is dependent on ice conditions, with constant route optimizations to avoid packed or ridged ice. Sea ice also guides the design of offshore structures, such as wind turbines, and in cold regions sea ice may be a hindering factor for renewable energy. In addition, large scale implementation of offshore wind farms may affect local sea ice dynamics. For all these purposes, new high resolution ice models capable of simulating ice dynamics across tens to hundreds of kilometers are needed.

Traditionally, large scale continuum models have been used for modelling sea ice dynamics on scales larger than kilometers. Such models are computationally efficient, and can easily be extended over larger areas and longer times compared to the

discrete element method (DEM) approach used here. A well known challenge with continuum models is that an effective rheology for sea ice has to be implemented in the model, and there is no easy and straight-forward way to model all relevant aspects of sea ice dynamics with a large scale effective rheology. Some of the early attempts in this direction was the visco-plastic model (Hibler, 1977; 1979) developed already in the 1970's. The visco-plastic model by Hibler can capture some large scale effective ice dynamics, but fails to model formation of leads, compression ridges, shear zones and floe fields that are obvious on scales smaller than $\sim 100 km$. More advanced and more accurate continuum models are, e.g., the elastic-decohesive model of Schreyer et al. (2006) and the Maxwell elasto-brittle model by Dansereau et al. (2016) and the brittle Bingham-Maxwell rheology model by Olasson et al. (2022). Several modern high resolution continuum models are able to capture many of the characteristics of large scale fracturing (Bouchat et al. (2022); Hutter et al. (2022)), and some are even utilized as operational applications tools with a few kilometers grid resolution (Pemberton et al. (2017); Kärnä et al. (2021); Röhrs et al. (2023)). However, even advanced and modern continuum models struggle with modelling fine-scale details such as leads and ridges.

DEM models take a significantly different approach. Instead of modeling sea ice as a continuum, solid and elastic blocks are initially connected together to form sea ice. The dynamics is typically computed via discrete versions of Newtons equations with some sort of energy dissipation terms. When load is applied, the connections may break, and ice disintegrates into discrete floes. Early models of this kind utilized circular Discrete Elements (DEs) moving in two dimensions (Babic et al. (1990); Hopkins and Hibler (1991); Blockley (2020)). Hopkins and Thorndike (2006) modeled Arctic pack ice using a DE-model. The resolution of these models were not enough to resolve details, instead important features, such as ridging, were described by an ice floe interaction model. A similar approach was later adopted by West et al. (2022) who simulated ice dynamics in the Nares Strait, and by Damsgaard (2021, 2018) investigating pressure ridging. Also a recent investigation by Manucharyan and Montemuro (2022), introducing complex discrete elements with time-evolving shapes, relied on a similar approach. In addition, they used a rudimentary fracture model to describe the failure of sea ice. Our model is not based on these models, instead we explicitly model ice dynamics at a scale in the order of meters, including ridging, leads, shear and tensile fractures, with the large-scale ice failure patterns emerging as collective results of these smaller scale failure processes. Neither does our model rely on an assumption of ice floes, but instead we let the ice floes form and fracture throughout the simulations.

The objective of this investigation is to bridge gaps between continuum models and DE-models by implementing and testing a computationally efficient DEM that has been developed and optimised for high-end computing with vast numbers of highly efficient processors. If a detailed high-fidelity model of this kind can be scaled up to length scales at which continuum models are sufficient and if the two can be combined into a unified framework, a very useful forecast model for sea ice dynamics would be the result.

## 2 The HiDEM model for sea ice

### 2.1 Mechanics of HiDEM

The HiDEM code uses a discrete element method (DEM) algorithm. A DEM formulation for sea ice motion may be related to the Cauchy momentum equation (Acheson (1990)), which treats sea ice as a continuum. In its full form, this equation accounts for coriolis force, atmospheric and ocean stresses, sea surface tilt and Cauchy stresses within pack ice. The Cauchy momentum equation reads

$$m\left(\frac{D\boldsymbol{u}}{Dt} + f\boldsymbol{k} \times \boldsymbol{u}\right) = \boldsymbol{\tau}^a + \boldsymbol{\tau}^w + m\boldsymbol{g}\Delta H + \boldsymbol{\nabla} \cdot \boldsymbol{\sigma}, \tag{1}$$

where $m$ is the combined ice and snow mass, $\boldsymbol{u}$ is the horizontal ice velocity vector, $f$ is the Coriolis parameter, $\boldsymbol{k}$ is the upward unit vector, $\boldsymbol{\tau}^a$ and $\boldsymbol{\tau}^w$ are the stresses due to air and water drag, $\boldsymbol{g}$ is the gravitational acceleration, $\Delta H$ is the vertical component of the sea surface tilt, and $\sigma$ is the Cauchy stress tensor of ice.

Below we focus on short term sea ice deformations driven by the external forcing and modified by the coastal boundary conditions and sea ice fracturing. In this case, we can neglect the coriolis term and the sea surface tilt, after which the previous equation becomes

$$m\frac{D\boldsymbol{u}}{Dt} = \boldsymbol{\tau}^a + \boldsymbol{\tau}^w + \nabla \cdot \boldsymbol{\sigma}. \tag{2}$$

Assuming a simple linear Kelvin-Voigt type of viscoelasticity (Meyers et al. (2009)) the stress tensor for sea ice can be written as,

$$\boldsymbol{\sigma} = A\dot{\boldsymbol{\epsilon}} + B(\boldsymbol{x},t)\boldsymbol{\epsilon}, \tag{3}$$

where $A$ represent dissipative deformations like viscosity, and $B(\boldsymbol{x},t)$ represent spatially and temporally varying brittle elasticity. Here, $\boldsymbol{\sigma}$ is stress, $\epsilon$ is strain and $\dot{\boldsymbol{\epsilon}}$ is strain rate.

DEM algorithms do not explicitly solve continuum equations, but instead resolve forces on interacting DEs. The continuum equations above can be reformulated in a format more suitable for a DEM implementation. DEs interact pairwise either through beams connecting them or through repulsive contact forces. If we define the discrete position vector, $x_i$, of the DE $i$, which include three translational and three rotational degrees of freedom, the DEM equations of motions can be written as

$$m_i\ddot{\boldsymbol{x}}_i + \sum_j K\boldsymbol{x}_{ij} + \sum_j C_2\dot{\boldsymbol{x}}_{ij} + C_1\dot{\boldsymbol{x}}_i = \boldsymbol{F}_i, \tag{4}$$

where $m_i$ is the mass or moment of inertia of $i$, $C_1$ is the drag coefficient of the combined drag of water and air, $\boldsymbol{F}_i$ are external forces and moments, such as gravity and buoyancy. Further, $K = K(t)$ and $C_2 = C(t)$ represent elements of contact stiffness and damping matrices for interacting discrete element pair $i$ and $j$ and $\boldsymbol{x}_{ij}$ refers to position vector between $i$ and its neighbors $j$. $\dot{x}$ and $\ddot{x}$ are first and second time derivatives of $x$. $K$ and $C_2$ correspond to B and A of Equation 3, respectively, and depending on the pair of discrete elements, they may include either elements of stiffness and damping matrices of the beams or those related to repulsive contacts of the discrete elements.

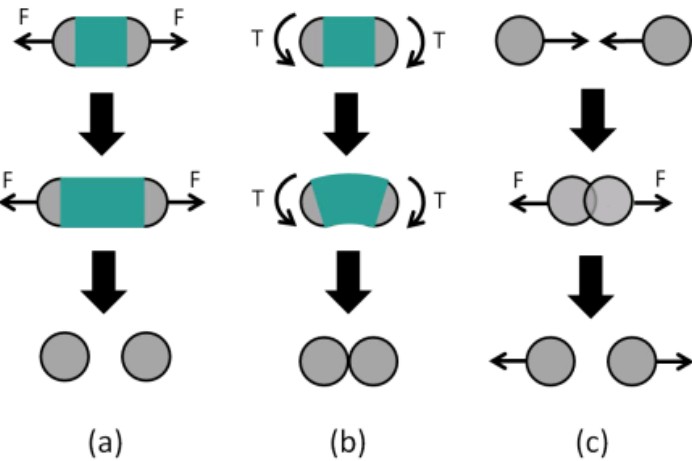

**Figure 1.** In a DEM algorithm, an intact material is described by joining discrete elements (here of circular shape) by beams: (a) Tensile forces (F) break a beam connecting DEs if stretched beyond a limit, (b) Torque (T) also breaks a beam if difference in rotation angles is too large. (c) Discrete elements also interact through pairwise in-elastic collisions (Riikilä et al. (2015)).

In the previous equation, $\sum_j K \boldsymbol{x}_{ij} + \sum_j C_2 \dot{\boldsymbol{x}}_{ij}$ corresponds to $\nabla \cdot \sigma$ in Equation 2, with the divergence operator being replaced by a sum over all of its neighbouring discrete elements of $i$. This can be done as the contact forces from neighbors on opposite sides of a discrete element cancel each other if they apply equal force on $i$ and, thus, only change in the force across an element induces motion. Further, $C_1 \dot{\boldsymbol{x}}_i$ and $\boldsymbol{F}_i$ of Equation 4 include the effect of stresses $\boldsymbol{\tau}^a$ and $\boldsymbol{\tau}^w$ in Equation 2. DEM simulations utilize explicit time stepping. For this, the previous equation can be written in discrete form by using the definition of derivatives, and the motion of the discrete elements, $\boldsymbol{x}(t+dt)$ as function of $\boldsymbol{x}(t)$ and $\boldsymbol{x}(t-dt)$, can be computed via iterations of time-steps ($dt$) based on element positions, velocities and forces acting on it.

## 2.2 Code optimization for High-Performance Computing

Any computational implementation of Eq. (4) is a trade-off between accuracy and computational efficiency. A higher accuracy would mean, e.g., including irregular elements, higher-order time-integration schemes, non-linear elasticity and/or non-linear drag coefficients. In contrast, simpler models with a higher computational efficiency allow for a finer resolution, i.e., smaller elements and timesteps. HiDEM is focused on the latter. In the DEM algorithm a large set of elements move relatively to each other and interact only with neighbors within a limited maximum interaction range. A clear majority of the computational effort for algorithms of this kind has to be dedicated to computing forces between elements. With such a code structure HiDEM is a good candidate for efficient implementation on the most powerful modern high performance computers (HPC). The HiDEM code is written in C++ with MPI (Message Passing Interface) and OpenMP (Open Multi-Processing, version 4.x or higher) for parallelization and multithreading. Offloading to GPUs is done using Cuda/Hip. The code is optimized for maximum

computational efficiency on supercomputers or large clusters with an efficient interconnect. The code can may be compiled for running only on CPUs or a combination of CPUs and GPUs with almost all computations taking place on the GPUs.

The HiDEM code has thus two levels of parallelization in two different ways: MPI message passing between CPU nodes, and OpenMP multithreading on CPUs with many compute cores, and, alternatively, MPI for CPUs and offloading the most compute intensive parts, using Cuda/Hip, to GPUs. This structure creates a high complexity of the code, and extreme care has been taken to implement optimal data structures and communications so that the compute power of the GPUs can be utilized as efficiently as possible. The technical details of the codes data structures and communication schemes are outside the scope of this article and will be reported elsewhere.

The results reported here were run on the LUMI supercomputer in Kajaani, Finland. In June 2023 LUMI was ranked third on the Top 500 list of the world's fastest supercomputers. The LUMI GPU-partition has 2928 GPU nodes each with a 64 core CPU and 8 Graphics Compute Dies (GCDs). For the results reported here we performed runs with about 100 million elements with roughly half a billion interactions, and a few million time steps. A simulation lasted typically about 10-20 hours, and we used no more than 4 GPU-nodes hence there is still a lot of potential to scale up both element count and to increase time integration speed. The extreme computational efficiency of the HiDEM code implemented in a suitable HPC environment, allowing for extreme scale and resolution properties, is what sets HiDEM apart from standard DEMs. DEM results cited in the Introduction above, typically report models with the order of 10,000 elements or for some larger element numbers, significantly larger timesteps for km-size elements. The resolution of the HiDEM simulations is demonstrated in Fig. 2B, which displays only 1% of the Kvarken simulation domain in order to make details visible. This figure also displays how damaged ice (i.e., drift ice) and undamaged (i.e., landfast ice) can behave differently in the model.

## 2.3 Sea ice simulations

The purpose of this investigation is to a apply a simple and computationally efficient DEM implementation, as described above, to simulate sea ice fragmentation and compare the result to observations. We apply the HiDEM code to ice failure in the Kvarken region of the Baltic Sea and to ridge formation in the Gulf of Riga (Fig. 2). Our objective is to investigate the model's capability to mimic the ice dynamics in Kvarken which is a narrow strait that is often ice covered on its NE side, while the sea often remain largely open on its SW side, creating interesting dynamics when wind presses ice towards the Southwest. Our second objective is to test how well the model mimics formation of ice compression ridges in the Gulf of Riga under strong SW winds. This is a well-known problem for shipping in that region.

We use close-packed spherical DEs, all of similar size, 8 meters in diameter, and connected by Euler-Bernoulli beams. A beam connect two center points of a DE. Each DE, and thereby also the endpoints of a beam, have 6 degrees of freedom: three translational, and three rotational. Beams connect all, or a fraction of randomly selected, nearest neighbors. The matrix $K$ in Eq. (4) contains the stiffness elements (or spring constants) that relates forces and torques to beam deformation. The stiffness matrix of a single beam, and other details, are given in Åström et al. (2013). All relations between forces and deformations are linear up to a beam breaking point, which is determined by the beam deformations, either as an elastic energy criterion or as a maximum stress/strain criterion. Once a beam is broken it vanishes. I.e., the connection between the DE's is irreversibly

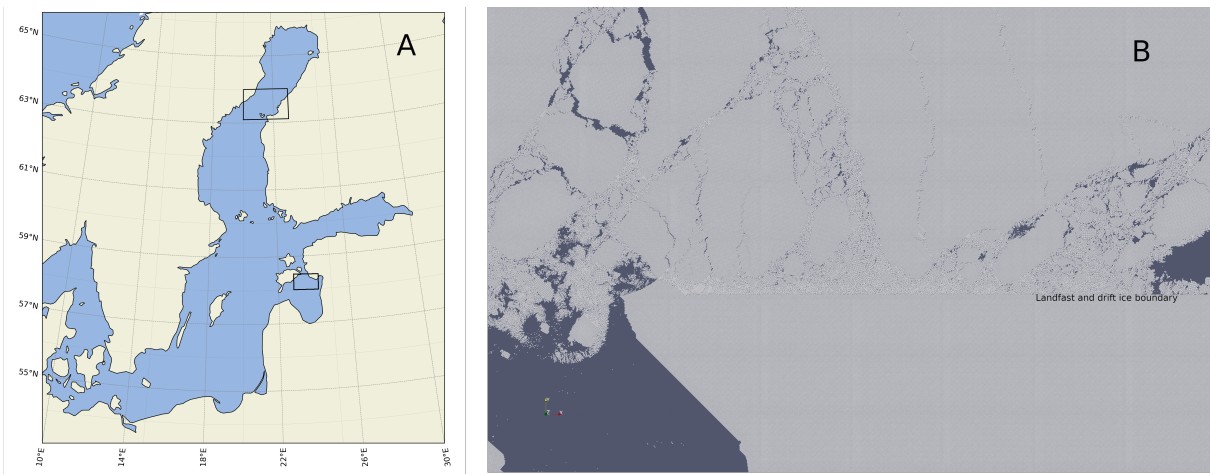

**Figure 2.** (*A*) The two simulation domains in Kvarken and the Gulf of Riga indicated by rectangles. (*B*) All DEs displayed in a $\sim 10km \times 7km$ area in the south-western corner of the Kvarken simulation domain at a late stage of the 8/3/2018 simulation when the ice is broken up. The straight boundary, from east to west, between drift and landfast ice is indicated in the figure.

**Table 1.** HiDEM parameters.

| | |
|---|---|
| Young's modulus | $2 \times 10^9 N/m^2$ |
| Fracture stress | $5 \times 10^{-4} N/m^2$ |
| Fracture mode | tension + constant*(bending and torsion) |
| Ice density | $910 kg/m^3$ |
| Water density | $1027 kg/m^3$ |
| DE diameter | $8m$ |
| Air drag | $1 Ns/m^3$ |
| Water drag | $20 Ns/m^3$ |
| Land friction | $10^6 Ns/m^3$ |
| Damping/critical-damping | $10^{-3}$ |

broken, and the DEs can freely move apart but will continue to interact if they are pressed against each other. DEM parameters are listed (Tab. 1). Drag and friction terms are linear in velocity. Drag coefficients are small to allow swift dynamics. Land friction is high to hinder ice from sliding on land. Damping is small compared to the critical damping of a harmonic oscillator to allow sound waves to travel in the ice, but large enough to hinder build up of vibrational kinetic energy in the ice. The element interactions are sketched (Fig. 1). The animation in the Supplementary Material demonstrate simulated ice dynamics in a small fraction ($\sim 0.3\%$) of the Kvarken domain so that details can be seen.

The typical winter sea-ice thickness in the Kvarken region is of the order of one meter, or less. It means an accurate ice thickness can only be described explicitly if the diameter of the spherical elements is no more than one meter. This would

increase computational requirements immensely compared to the 8-meter spheres we use for the large scale simulations below. The number of elements would have to be increased by a factor $8^2$ to simulate the same domain. Instead, we use a single layer of DEs in a close-packed configuration forming a triangular lattice of 8 meter spheres.

A consequence of the 8 meter diameter is that the model ice will be significantly thicker and stronger than the ice that appears naturally in the region. Ice fractures when stress build-up reaches the fracture threshold of the ice. When the ice breaks, stress is relaxed. In order to simulate this we can define a dimensionless parameter, $R_{ss}$, that is the ratio of stress to strength of the model ice, and tune the applied stress on the ice in the simulations so that that this ratio is approximately equal to unity. The stress-to-strength ratio of the model ice is given by

$$R_{ss} = \frac{h \, l_{DE} \, E_{ice} \, \epsilon_{frac}}{f_{DE} \, L_{domain}/l_{DE}}, \tag{5}$$

where $h$ is ice thickness, $l_{DE}$ is the horizontal dimension of the DEs, $E_{ice}\epsilon_{frac}$ is the ice fracture stress, $f_{DE}$ is the force applied on each DE, and $L_{domain}/l_{DE}$ is the relative resolution of the simulation domain. $R_{ss}$ is of order one, as it should be, if we use: $h = 1$, $l_{DE} = 8$, a driving forces of the order of $100N$/DE, an ice fracture stress of order $10^5 N/m^2$, and $L_{domain}/l_{DE} \sim 10^4$. A benefit of increasing $f_{DE}$, keeping $R_{ss}$ fixed, is that ice dynamics can be made a bit faster, and forecasts corresponding to longer times can be done with shorter simulations.

The triangular lattice structure introduces a weak anisotropy in the material stiffness and limits the crack propagation directions to a few preferred ones on the scale of a DE. The triangular lattice has three possible crack propagation directions with a 60 degrees angle between them. These angles are however not visible in the larger scale fracture patterns in e.g. Fig. 2B, which means, on a large scale the model behave predominantly isotropic, as it should.

In spite of the limitations, the lack of details in the initial and boundary conditions, driving forces, and the simplicity of the DEM implementation, the model is, as demonstrated below, able to capture a great deal of the large scale structures and small scale details of observed sea ice fragmentation and dynamics.

## 3   Kvarken region March 2018

Kvarken is the narrow and shallow neck between the Bay of Bothnia and the rest of the Gulf of Bothnia. In a typical winter, such as the winter 2018 was, the Bay of Bothnia freezes over completely, while the rest of the Gulf of Bothnia freezes only partly. This makes Kvarken an interesting location for ice dynamics as, with strong Northern or Eastern winds, the sea ice in the Bay of Bothnia is fragmented and pushed through the narrow Kvarken Strait. During severe winters, ice arches can develop on the Northern side of Kvarken. Physically, ice dynamics in Kvarken resembles that of Nares Strait where ice arching is common Moore et al. (2023).

We simulate two different cases of ice dynamics, which occurred on 8 and 23 March 2018. For the simulation domain we use a $\sim 100$ m resolution digital depth model of the Baltic Sea (courtesy of Baltic Sea Hydrographic Commission), and for comparison with simulation results we use Copernicus satellite images from the LandSat program (see data availability below). For forces driving the ice fragmentation we mimic wind directions and magnitudes from weather data archives.

Initially ice is set to cover the entire domain, except for a region south-west of the narrowest part of Kvarken, where we initially have a rectangular area of open water to roughly mimic the ice situation in March 2018. The northern and eastern domain boundaries are fixed, while the southern and western boundaries allow ice to flow out of the domain, except where land is blocking ice motion, obviously. Discrete element diameter is 8 m, and we set the beam width to $40\%$ of the diameter. Further, we introduce disorder and strength variations in the ice by initially reducing the density of beams from its maximum,

at uniformly random and uncorrelated locations. We use slightly different setups for the two cases: The density of the beams is reduced by $40\%$ over the entire domain (23/3/2018), and for partly refrozen ice rubble that often appear at open sea we reduce the density of beams by $40\%$, while fast ice regions in the inner archipelago has zero reduction in beam density (8/3/2018). In terms of ice strength this corresponds to ice that is about 1.3 m thick.

### 3.1   Simulation case study for 23rd March 2018

For 23/3/2018, nearby weather stations reported moderate western wind early on the 22 of March, which then strengthened and turned to Northern wind $9-11m/s$, then turned to North-Eastern, and eventually weakened during 23 of March. To model this we applied a constant force, from the north, on all elements for 3 hours, followed by 45 minutes of force from the north-east. In this case, as explained above, the ice is similar over the entire domain. Fig. 3A displays the resulting ice motion, while Fig. 3B shows the largest compressive strains at the end of the simulation. These two figures show clearly the 'bottleneck

behaviour' of ice motion through the Kvarken Strait. Ice that have passed southwest of the narrowest region move fast, while compressive stresses are built up northeast of the narrowest region leading to reduction of ice speed. Ice is breaking up along compressive shear fracture zones, and some evidences of compressive arches are visible upwind of the narrowest point of the strait. Additional ice compression, that is not related to ice motion through the strait, is visible at the northern side of the Finnish archipelago.

Figure 3C depicts satellite image of the Kvarken on 23 of March. Correspondingly, Fig. 3D shows simulated fracture pattern. The model derived figure is rendered to mimic the satellite image except for land that is brown so it can easily be distinguished from open water. The similarity between these two images is striking, but there are also some noticeable inconsistencies: *(i)* There is significantly more open water east and north of Holmön (marked by 'H' in Fig. 3A). This is most likely due to differences in initial conditions. In the simulations the initial condition was a $100\%$ ice covered Bay of Bothnia, while in the

reality there existed a wide lead of open water along the Swedish coast the days before the 23 of March. *(ii)* Ice floes has travelled much further through the Kvarken Strait in the satellite image compared to the simulations for which the floes are closer to their original position. The reason for this is simply that the simulation cover 3 hours and 45 minutes of ice motion, while in reality the motion has lasted for about a day.

Figures 4A and 4B highlight the drift and the landfast ice in the satellite and simulated images, respectively. Also in this

case are the similarities between the two images striking, but there are also visible differences: The fairly straight south-west north-east lead that marks the boundary between drift and landfast ice goes a bit more to the north in the simulation compared to the satellite image. This lead begins near Valsörarna (marked by 'V' in Fig. 4B) and reach the Finnish coast at the Öuran Island (marked by 'Ö') in the satellite image, and further north near Torsön Island (marked by 'T') in the simulations.

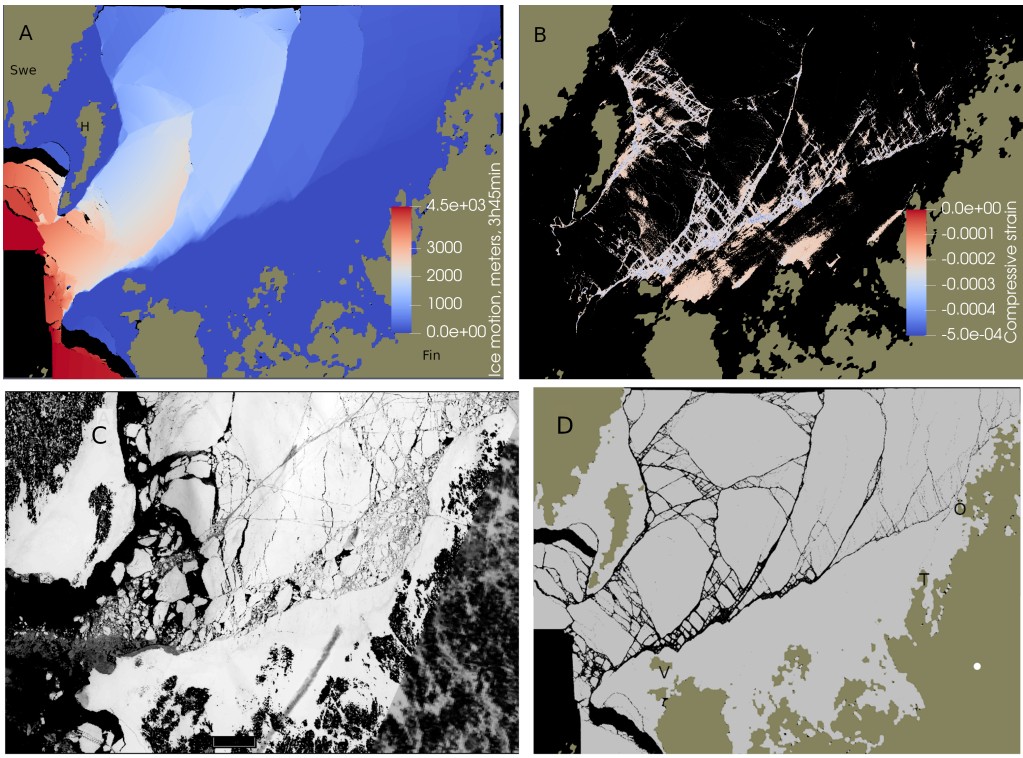

**Figure 3.** (*A*) Color coded ice motion for the 23/3/2018 simulation. 'Swe' and 'Fin' marks the Swedish and Finnish mainlands. 'H' marks the location of Holmön. (*B*) The largest compressive strains on intact beams connecting DEs at the end of the simulation. (*C*) A satellite image of the Kvarken area on 23/3/2018. 'T' marks the location of the Torsön Island, 'Ö' the location of the Öuran Island, and 'V' the Valsörarna Island. (*D*) The simulated fracture pattern after 3 hours and 45 minutes. This image display (with black dots) all beams that are strained more than $5\%$ of their original length (and thereby obviously broken). Water is black, ice is gray, and land brown.

Figures 4C and 4D highlight regions with highly disintegrated ice adjacent to the boundary between drift ice and landfast
ice. The reason why the ice become so much crushed in this region is that it is pressed southwards by the wind, and high
compressive forces will therefore appear on the northern side of the Finnish archipelago. The difference in shape and extent
of the simulated and observed crushed ice regions are again a consequence of the shorter ice dynamic time in the simulations.
A longer simulation would induce more shear crushing against the fast ice margin as the drift ice slowly move west through
the Kvarken Strait. Another consequence of this particular dynamics is appearance of east-west tensile stress in the drift ice
region. Such stresses are typical for shear zones and often induce tensile cracks, more or less, perpendicular to the shear zone.
Such cracks are marked by 'C' in Figs 4C and 4D.

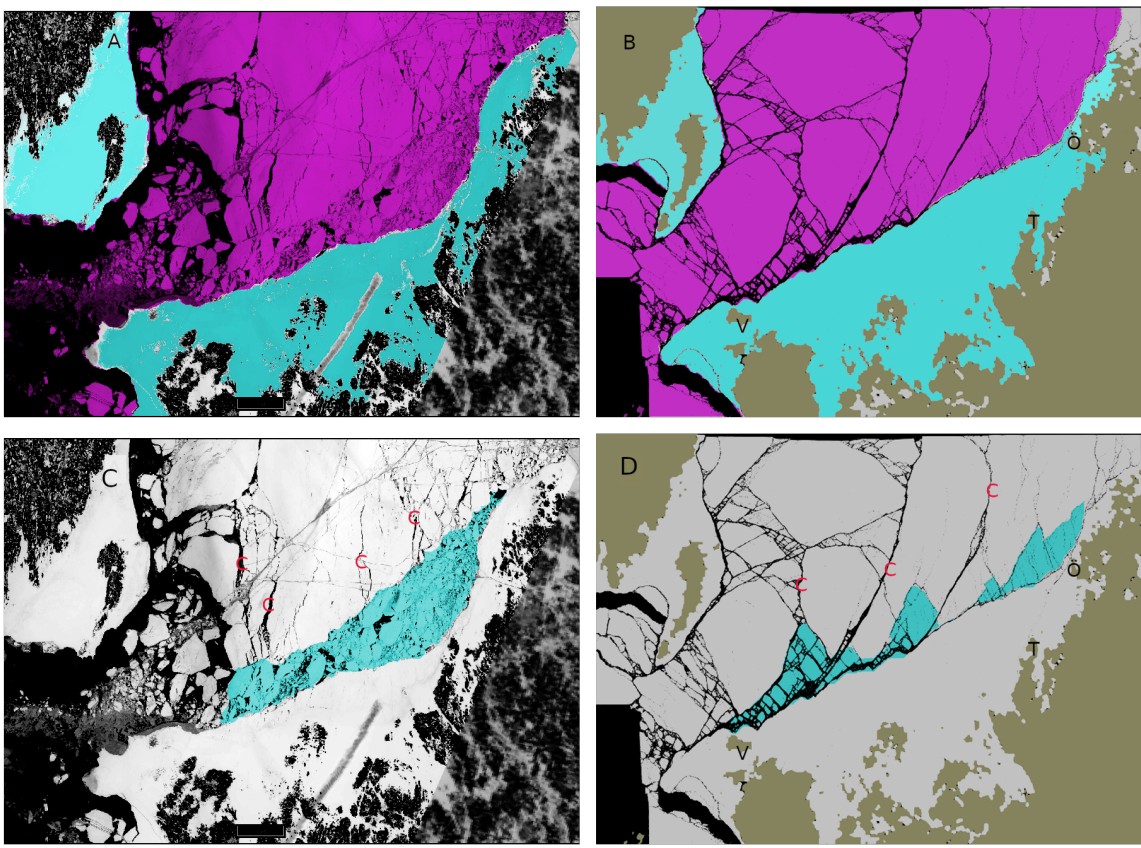

**Figure 4.** (*A*) Fast ice (teal) and drift ice (purple) regions extracted from the satellite image for 23/3/2018. (*B*) Fast and drift ice regions at the end of the simulation. (*C*) Crushed region and tensile cracks marked by 'C' extracted from the satellite image. (*D*) Crushed regions and tensile cracks at the end of the 23/3/2018 simulation.

### 3.2 Simulation case study for 8th March 2018

The other date for model testing in the Kvarken region is 8/3/2018. During a few days proceeding this day there was a fairly constant mostly eastern wind. We use the same initial state in this case as for 23/3/2018, except that we now define two region of stronger landfast ice to test how this influence the outcome of simulations. One region of stronger ice is the strait between Holmön and the Swedish main land, and the other region is the Finnish archipelago along the southern border of the domain terminating close to Valsörarna (marked by 'V' in Fig. 4D). The effect this has is visible, for example, in Fig. 2B. The damaged ice breaks, while the fast ice remain solid. Another differences to the previous case is that now the wind-stress forcing on DEs comes from the east and not the north, and the simulation is a bit shorter (3 hours 15 minutes instead of 3 hours 45 minutes).

Figure 5A display Ice motion, while Fig. 5B shows the largest compressive strains at the end of the simulation. Similarly to the 23/3/2018 case, ice is pressed through the Kvarken Strait, but now, because of a different wind-stress direction, the ice

comes more from the east than from the north. Compressive stress builds up northeast of the strait as in the previous case, but in this case there is also significant compressive fracture of ice against the Swedish coast (Fig. 5B). The ice begins to break up along an east-west corridor ending between the south-end of Holmön and the Finnish archipelago. The same corridor of fragmented ice can be seen in the satellite image, but in this cases it reaches almost all the way to the Finnish coast. It is quite clear that the simulation would have to run longer for significant fragmentation to reach that far east, even though the ice forcing is slightly exaggerated in the simulations, as explained above.

Figure 5C displays a satellite image of the region on 8/3/2018. In contrast to the previous case, the simulation image rendered to mimic the observations now displays, as before, fractured beams as black to have the same color as open water, but on top of them all intact compressed beams are rendered as light gray to mimic regions in the satellite image that may have densely packed drift ice and would therefore appear white or grayish. It is not possible to determine from the satellite image (Figure 5C) if the regions near the Swedish coast, northeast of Holmön, is densely packed drift, fast ice or a mix of both. This issue may have two explanations: Either the crushing of ice cannot be detected in the satellite image as it does not expose dark open water, or it may be that the wind forcing was set too strong in the simulations. The latter is consistent with the weak to moderate eastern winds during 6 to 8 of March. Inspecting satellite images for 6 and 7 of March reveal that it took at least 3 days to form the east-west fragmentation corridor. With the current model configuration it would take a lot of computational resources to execute simulations for such durations.

Figures 5E and 5F highlight (teal coloured) the floes that are about to flow out through the Kvarken Strait. The same figures display the land and landfast ice on the eastern side of Kvarken as purple areas. The teal coloured areas display obvious similarity, and the eastern border between drift ice and landfast ice, indicated by the boundary of the purple areas, are very similar for the observed and the simulated. A dominant diagonal lead, marked by 'Dia' appears in both images. The exact locations of this lead differs however a bit between the observed and the simulated. Finally, marked by 'Arc' there are cracks in both images that have the characteristic curved shape of cracks formed in regions with compression arches.

### 3.3 Floe size distributions

The floe size distribution (FSD) that form in the simulations may be extracted. We were not, however, able to extract the corresponding FSDs from the satellite images. Observed FSDs have recently been published for the Canada Basin (Denton and Timmermans (2022)). They reported power-law FSDs, $n(s) \propto s^{-\alpha}$, with exponents $\alpha$ ranging from 1.65 to 2.03 over a floe-area range from $50m^2$ to $5km^2$, with the larger exponent values appearing in the summer and autumn, and at low sea ice concentration. Figures 6A and 6B display the FSDs from the 8/3/2018 and 23/3/2018 simulations, respectively. Power-laws are evident with exponents 1.72, and 1.76, respectively. Also the size-ranges are similar to those reported by Denton and Timmermans (2022). However, the discreteness of the DEM become influential for the smallest floe sizes: a single element have an area of $\pi r^2 \approx 50m^2$. As single DEs cannot be broken there is a 'pile up' effect in the FSDs for floes with a single or a few DEs. The largest 'floes' in the FSDs, outside of the power-law range, represent the fast ice regions.

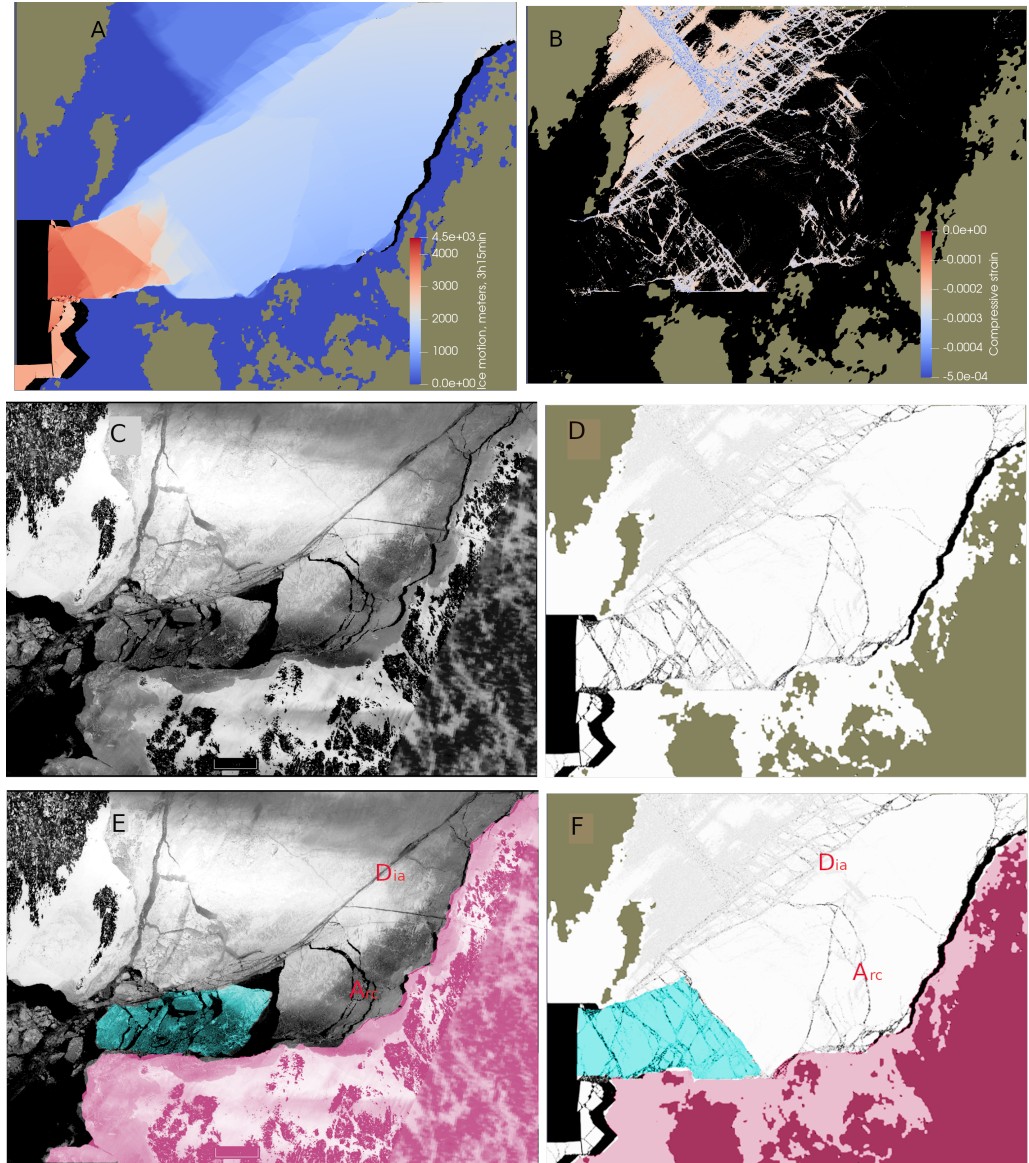

**Figure 5.** (*A*) Color coded ice motion for the 8/3/2018 simulation. (*B*) The largest compressive strains on intact beams connecting DEs at the end of the simulation. *C*) A satellite image of the Kvarken area on 8/3/2018. (*D*) The simulated fracture pattern after 3 hours and 15 minutes. This image display (with black dots) all beams that are strained more than $5\%$ of their original length (and thereby obviously broken), and beams compressed more $0.02\%$ (light gray dots) of their original length. (*E*) Highlighted (teal) drift ice that is on its way out through the Kvarken Strait. Purple area cover the land and landfast ice on the eastern (Finnish) side of Kvarken to highlight the boundary between drift and fast ice. (*F*) Corresponding highlighted regions for the simulation. 'Dia' marks the dominante diagonal lead, and 'Arc' marks cracks formed in regions with compression arches.

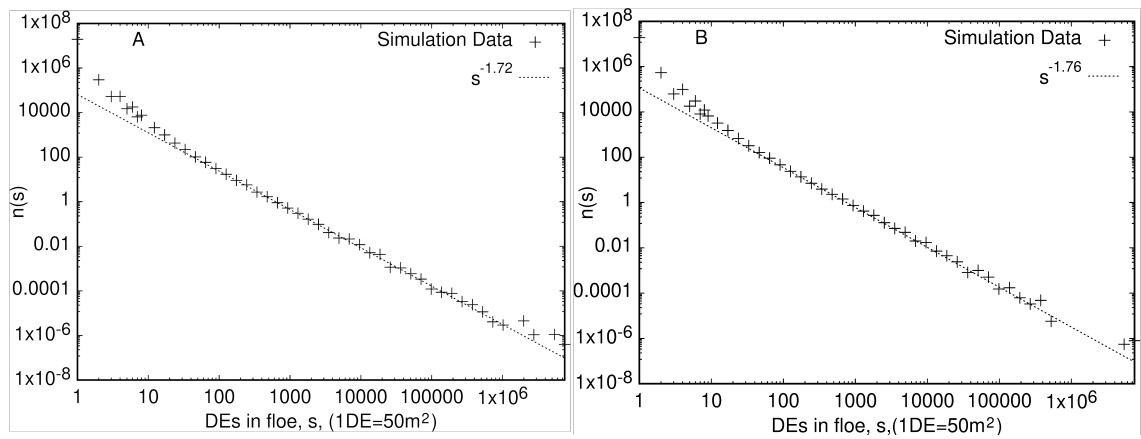

**Figure 6.** (*A*) The computed FSD at the end of the 8/3/2018 simulation fitted by power-law with exponent 1.72. (*B*) The computed FSD at the end of the 23/3/18 simulation fitted by power-law with exponent 1.76. The scale on the x-axis is number of elements, DEs, in a floe (1 DE$\approx 50m^2$)

## 4    Gulf of Riga

An important characteristic of sea ice compression is the formation of pressure ridges. In order to demonstrate how pressure ridges form in HiDEM, a square shaped sea-ice sheet of size 10 km $\times$ 10 km was modelled using DEs of 1m diameter and subjected to uniaxial compression (Fig. (7)A). Our simulation results may be assessed using an aerial photograph of ice ridges in the Gulf of Bothnia from March 2011 (Fig. (7)B). The dynamics process of ridge formation become rather evident in these two images. Compression, induced by a strong wind, breaks up the ice in floes. Along the floe boundaries the ice fractures in

compressive shear zones and ice rubble builds up to form ridges. In the simulated image, the floes still remain largely at their original position in relation to each other, while they have moved enough to form patches of open water between them (Fig. (7B)).

    Pressure ridge formation is a particular hindarance for shipping in Baltic Sea. In the Gulf of Riga, ridges typically form under compression from southwestern winds. Such conditions are known to produce ridges, in particular between the Kihnu

and Saaremaa islands. Figure 8A shows the strains between DEs, that were initially connected by beams, at the end of a simulation. Both intact and broken beams are included. Formed compression ridges appear in the figure as vague red bands of tension in an otherwise compressive ice landscape. Figure 8B shows the FSD extracted from the Gulf of Riga simulation. The exponent in this case is significantly larger ($\alpha \approx 2.12$) compared to the Kvarken simulations. This is consistent with the topography of the Gulf of Riga not allowing the ice to flow out of the domain in contrast to the Kvarken Strait, and therefore

the ice floes are crushed and grinded to smaller sizes (Sulak et al. (2017); Åström et al (2021)). The strain-rate distribution can be extracted from the simulations. For the largest strains the distribution of rates (Fig. 8C) is consistent with power-law distributions observed at much larger scales (Girard et al. (2009)).

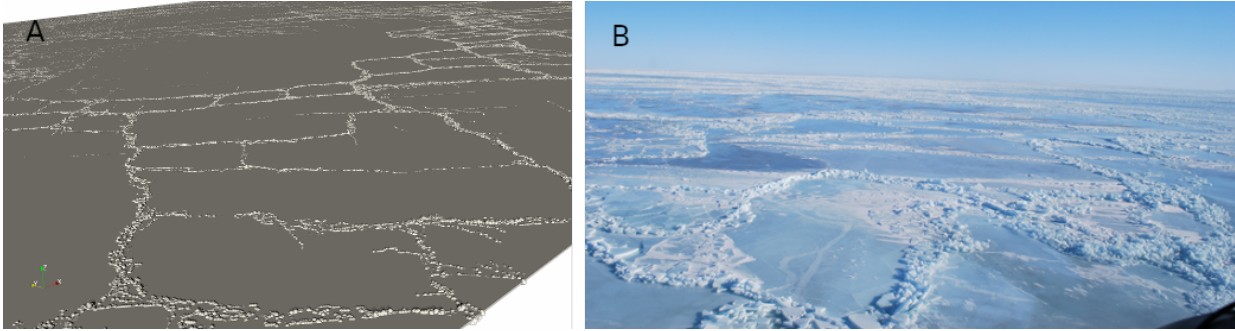

**Figure 7.** (a) HiDEM simulation of a failure of a 10 km × 10 km square-shaped sea-ice cover subjected to uniaxial compression. The sea-ice cover was modelled by using dense-packed single-layer $1m$-diameter spherical discrete elements. The sea-ice cover fails in shear and ice ridges are formed. (b) An aerial photo at $30m$ altitude of fractured ice with pressure ridges in the Gulf of Bothnia after a storm in March 2011 (J. Haapala , private photo).

To further investigate ridge formation in the Gulf of Riga we identify locations of compression ridges in the simulations as places where elements are pressed below the sea surface to form ridge keels. Ridges are affected by the bathymetry (Fig. 9A).
It is evident from this figure that when long ridges are formed in a single event, like in our simulations, the structure of the ridges is strongly influenced by the shape of the coast line and the bathymetry in shallow waters where ridge keels begin to get grounded. It is therefore reasonable to expect that ridge patterns form fractals, just like coastlines and many structures formed by sea ice dynamics do (Weiss (2001)). A simple box counting algorithm, $N(L/l) \propto L/l^D$ can reveal the fractal dimension $D$. Here $L/l$ is the linear number of boxes the domain is divided into, and $N$ is the number of boxes containing DEs identified
as ridge keels. Fig. 9B shows the result of this exercise, which indicate that $D \approx 1.12$, which is a fairly low dimension. $D = 1$ would mean that ridges form non-fractal linear structures. It is reasonable to expect that if ridge fields were formed over longer periods and by different wind directions they could eventually cover entire areas, and their dimension would then become $D = 2$. The fractal dimension $D = 1.12$ is a rather typical value for reasonably straight coastlines like in the Gulf of Riga.

Fig. 9C shows locations of ice ridges observed from ice charts during 2000-2016. The ridge locations follow reasonably well
the general ridge pattern of the simulations indicated by the reddish area in Fig. 9C. Fig. 9D shows the wind statistics (i.e. a wind rose) from the ERA5 dataset (location 58,00 and 23.75) for the same time (December 15th until May 1st in 2000-2016). This figure demonstrate that ridges are predominantly caused by SW winds, which is the dominant direction of strong winds in the area.

## 5 Discussion

The outcomes presented in conjunction with prior findings on ice shelf disintegration (Benn et al. (2022)), laboratory-scale ice crushing experiments (Prasanna et al. (2022)), and glacier calving (Åström et al (2021)) illustrate that HiDEM has the capability to model the physics of ice fragmentation. In practical terms, the current version of the code can be utilized to forecast

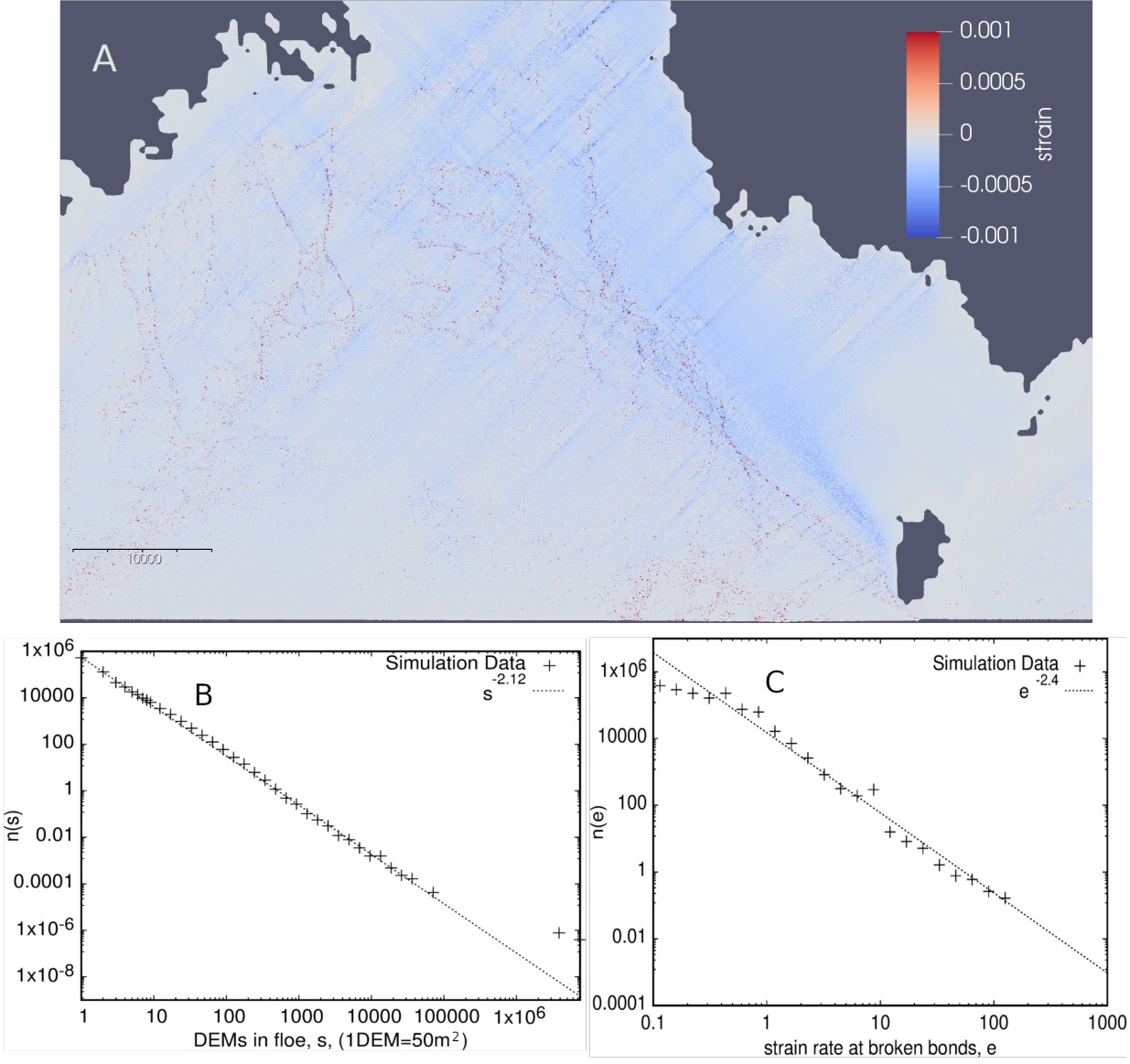

**Figure 8.** (*A*) Simulated strain field in the Gulf of Riga induced by South Western winds. (*B*) FSD of the Gulf of Riga simulation. (*C*) Strain rate distribution, $n(e)$, of the largest strains, $e$, in the Gulf of Riga simulation.

sea ice movement and fragmentation for a few days across distances of a few hundred kilometers. However, ensuring a consistent supply of accurate forecasts would necessitate a method to acquire a high quality initial conditions. This would not only require a comprehensive understanding of variations in ice thickness but also in ice quality. Additionally, accounting for spatial

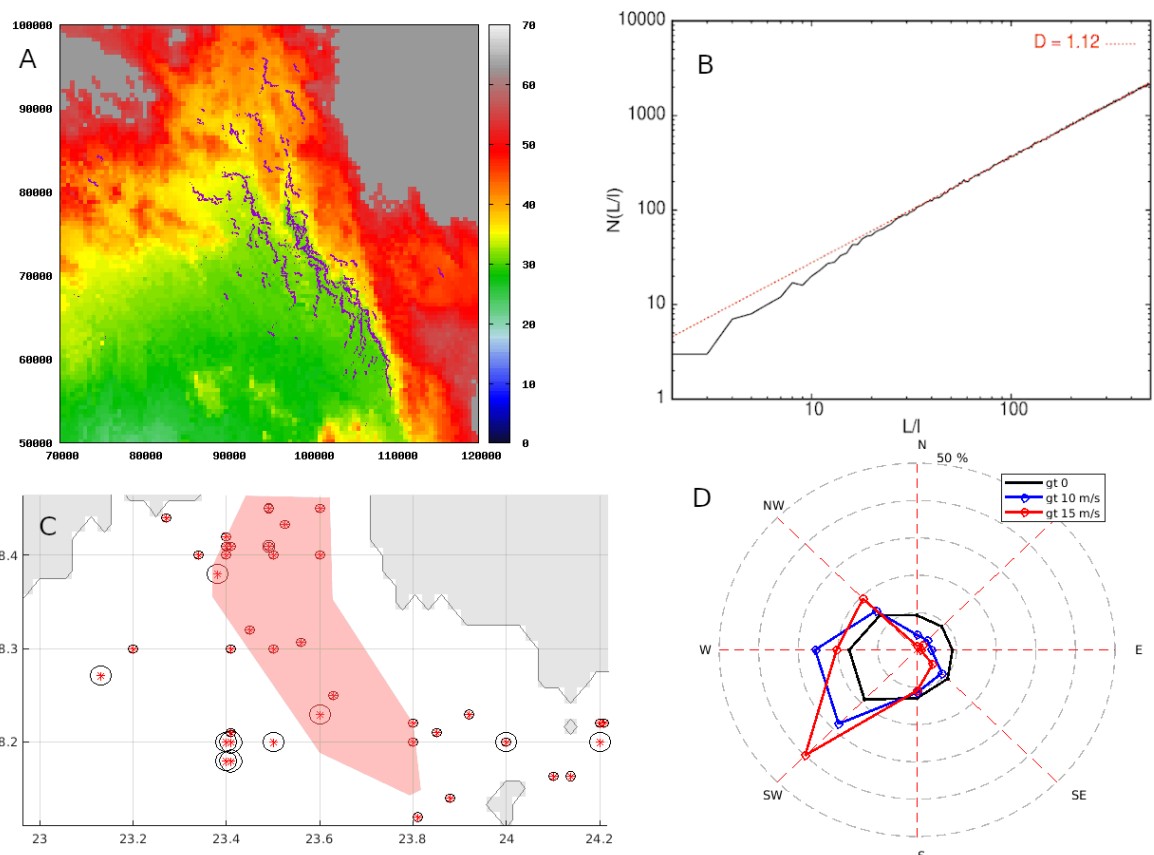

**Figure 9.** (*A*) Color coded bathymetry. The water surface is at 57m. Locations of DEs that make up compression ridges are indicated by blue markers. Single DEs on the surface would become grounded in dark red areas. Gray is land. Axis are in meters. (*B*) Result of a box-counting algorithm for compression ridges, $N(L/l) \propto L/l^D$, where $l$ is box length, $L$ is domain length, and $N$ is the number of boxes containing ridges. (*C*) Observed locations (lon,lat) of ridges from ice charts. The statistics is based on the data from winters 2000-2016 (15 December - 1 May). Red stars are ridge locations, and the diameter of the black circles indicates the number of days of ridged ice at that location. The reddish region outline the area where ridges form in the simulations (*D*) A windrose of winds greater than 0, 10 and 15 m/s for December 15th until May 1st, 2000-2016. Frequency intervals for the dashed lines is 10%.

variations in ice surface roughness is crucial as it impacts the local stress on the ice induced by wind and currents. Moreover, precise forecasts of winds and currents would be essential to determine the forces acting on the ice during simulations. Proper boundary conditions would also need to be established for each scenario, especially if the ice is permitted to exit the simulation domain. While evaluating this in cases where the domain is bounded by land is straightforward, it becomes more challenging 305 when the boundary crosses water with dynamic ice both inside and outside the domain. Enhancing the HiDEM code for sea ice forecasts would significantly benefit from improved comparisons with quantitative observational data on sea ice dynamics that could be directly juxtaposed with simulation data for any specific location. For instance, having detailed data on the evolving

floe size distribution, shapes, and locations in the Kvarken Strait for a specific timeframe would be highly valuable for further validating HiDEM. Similarly, recording the formation of compression ridges and ice motion in the Gulf of Riga, such as during a midwinter storm with southwest winds, would be equally beneficial. Any observed distribution function, velocity field, or stress field of this nature could be valuable for comparison with simulation results, provided that the simulation starts with precise initial conditions and the simulated ice dynamics is driven by valid forces.

One challenge is the disparity between the short time-step required for accurate fracture dynamics and the relevant timescales for sea ice dynamics. The time resolution, $dt$, must be smaller than the time it takes for sound to travel across a DE, which for ice translates to $dt < DE_{diameter}/\sqrt{K + 3G/4} \approx 0.0025 sec$, for the setup used in this investigation. $K$ represents the bulk modulus and $G$ is the shear modulus for ice. Here, a time step of $dt = 0.001 sec$ has been used. This implies that 3.6 million time steps are necessary to simulate 1 hour of ice dynamics. Even with the highly optimized HiDEM code, practical simulation times are typically constrained to a range from a few hours to several tens of hours, depending on the available computational resources. In contrast, the relevant timescale for sea ice dynamics may span days, weeks, or months. Although it is feasible to accelerate ice dynamics slightly in simulations compared to natural rates, the limitation in simulation times means that it is not feasible to compute entire winter seasons of sea ice dynamics. Instead, a snapshot of sea ice at a specific time must be generated based on observations, and the near-future ice dynamics can then be simulated from such a starting point.

It is important to note that the HiDEM code solely models sea ice dynamics as elastic-brittle fracture and dynamics, omitting the thermodynamic processes involved in sea ice formation and disintegration. Over extended periods and in extreme temperature conditions, thermodynamic processes often dominate sea ice behavior, while in shorter timeframes where ice is subjected to stresses surpassing its strength, elastic-brittle behavior typically prevails. A potential approach to encompass the full spectrum of processes would be to integrate a code like HiDEM with a large-scale continuum model that include the modeling of thermodynamic processes.

The ice breakup events in Kvarken in March 2018, as detailed in this report, were accompanied by air temperatures hovering around -10 ° C, indicating the possibility of new ice formation. Nevertheless, the influence of the windy weather conditions would have constrained freezing rather effectively during the timeframe of a few dozen hours pertinent to the modeled and observed elastic-brittle breakup.

## 6 Conclusions

In this study, we have utilized the HiDEM model to analyze sea ice fragmentation and have shown its capability to accurately replicate observed characteristics. Specifically, we have compared the outcomes of the simulations with satellite imagery from the Kvarken area of the Baltic Sea in March 2018. The external forces acting on the ice in the simulations were derived from weather archives. The fracturing of the ice and its movement through the narrow Kvarken Strait were primarily influenced by moderate to strong winds blowing from the North and East. Despite using an 8m grid resolution, minimal model adjustments, and basic initial conditions, the model successfully replicated a significant portion of the fracture patterns, fast ice distributions, and ice drift patterns observed in the satellite images. Furthermore, we explored the formation of compression ridges in the

Gulf of Riga and discovered that the size distributions of floes and the development of compression ridges aligned well with real-world observations. While the model offers detailed insights into fracture patterns, leads, compression ridges, and floes, its practical utility as a forecasting tool is constrained by certain limitations. The model's ability to simulate at a high resolution is restricted to relatively small domains, and the duration of simulated ice dynamics is also constrained. The most favorable method for improving the precision of ice dynamics predictions seems to be a blend of DEM and continuum models, as these two model types possess contrasting strengths and weaknesses.

*Code and data availability.*  A release version of HiDEM is available at https://doi.org/10.5281/zenodo.1252379. The bathymetry for the simulations are provided by the Baltic Sea Hydrographic Commission freely available at: http://data.bshc.pro. The satellite images are: ESA Copernicus Sentinel Data SYKE (2018), and USGS/NASA Landsat program SYKE (2018), provided by the Finnish Environment Institute SYKE, available at: https://wwwi4.ymparisto.fi/i4/eng/tarkka. Supplementary Material at https://zenodo.org/records/10471034.

*Author contributions.*  The authors FR and JÅ have constructed the HiDEM code. JÅ set up the simulations, performed them, analysed the results and has written parts of the paper. JH and AP contributed to analysis of the model results and writing of the manuscript. RU and IM contributed the observational data for the Gulf Riga and participated in writing of the manuscript.

*Competing interests.*  At least one of the (co-)authors is a member of the editorial board of The Cryosphere.

*Acknowledgements.*  This work was supported by the NOCOS DT project funded by the Nordic Council of Ministers. JH and AP wish to acknowledge funding from the European Union – NextGenerationEU instrument through Academy of Finland under grant number (348586) WindySea - Modelling engine to design, assess environmental impacts, and operate wind farms for ice-covered waters. We acknowledge constructive criticism provided by the referees.

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
