# Peer review of "A large-scale high-resolution numerical model for sea-ice fragmentation dynamics"

_The Cryosphere, 2023_

## Author Response (AR1)

Reply to Reviewer 1:

We would first like to thank the reviewer for useful and constructive criticism of our manuscript. A 'latexdiff' version of our updated manuscript with new figures attached at the end is available as a Supplement. The three main points raised by the reviewer are addressed below: (1) What sets our model apart from other DEMs? 2) The manuscript would benefit from more comparison with observational data. 3) Clarify the impact thermodynamic processes may have on the differences between observed and simualted ice dynamics. Finally, we give responses to specific points raised.

Reply:
1) We have now in much more detail explained what sets our code apart from other Discrete Element Models. In the new Section 2.2 we describe how the code have been optimized for, and how we utilize, high-end supercomputers. With this approach we have been able to scale down both time and space resolution several orders of magnitude smaller than standard DEM implementations. Published papers of DEM's for sea ice do not typically list both their space and time resolution but in the cases they do, elements used are often of the order of 10,000 or less (West et al. (2022), Damsgaard (2021), Manucharyan and Montemuro (2022)). This is in sharp contrast to our roughly 100 million elements. Other published DEM's use a significantly different algorithm (like e.g. the 2D model of densly packed km-scale floes by Hopkins and Thorndike) that allows for much larger timesteps (about 1000 times bigger than ours). Even though we use several orders of magnitude smaller timesteps to resolve small scale features of ice dynamics, the computational efficiency of the code still allow us to use element numbers that are larger than that of H&T by a couple of orders of magnitude. In the current manuscript we use of the order of a few million timesteps. These simulations still utilize only a small fraction of the available supercomputers capacity. The extreme scaling of HiDEM allow us to compute ice dynamics on the scale of 100 km X 100 km, and still resolve ridge and lead formation on the 10 meter scale. We have now tried to explain this better in the text, in Section 2.2, in particular.

2) The reviewer correctly points out that our manuscipt would greatly benefit from more quantitative comparisons with observations/remote-sensing data. This is something we tried to achieve from the very beginning of this investigation. Even though this work was done within a large consortium with several leading institutions for sea ice observations in the Balctic sea (like Met offices of the Nordic countries, https://www.csc.fi/-/nocos-dt), we did not manage to get much useful data for comparison with the simulations results, simply because such data do not exist for the Baltic sea (in contrast to e.g. existing satellite remote sensing data for the central Arctic Ocean).

We have, however, considerably expanded the text and added images to improve the sections on the model results. The quality of the satellite images does not really allow for quantitative comparisons. Apart from the floe size distributions, we have therefore tried to get as much as possible of qualitative comparisons between simulations and the available satellite images that are now included in the manuscript. We have e.g. remade the figures that compares simulated and observed results, added several new panels, and considerably expanded the text: Now in Sections 3.2, 3.2 and 3.3.

For the ridge formation we have been able to obtain some, albeit low-resolution, observational data of locations of compression ridge formation in the Gulf of Riga 2000 - 2016. We have now added a comparison of this data to locations of ridge-formations in the simulations. For the Gulf of Riga we have furthermore reported the strain-rate distribution, the floe size distributions, and computed the fractal dimension of the ridges, and discussed its connection to the shape of the coast line and bathymetry.

3) We have now added to the discussion in section 5 text on how thermodynamic processes could explain differences between observed and simulated results.
* * *
Reviewer:

(1) Line 28-29. Give some references here as the examples of model failures.

Reply:

This refers to the visco-plastic model of Hibler (cited on the previous text row). This has now been made more clear in text.

Modified text:

"The visco-plastic model by Hibler can capture some..."
* * *
Reviewer:

(2) Line 35-45. There are lots of applications of DEM in sea ice modeling, but the discriptions here don't make me very clear on the difference between the previous DEM model and the present model.

Reply:

The difference between previous DEMs and the present model is the extreme scaling achived via an efficient High-Performance-Computing implementation, and the use of supercomputers. This is now described in section 2.2
* * *
Reviewer:

(3) LIne 72. The abbreviation DEs has already shown on line 38.

Reply:

This is now fixed.
* * *
Reviewer:

(4) What is the purpose of Fig 2B ? To compare the simulation result with in-situ conditions only by photos? They are different in many aspects, and what we can say from Fig.2a is just we can produce ice ridges by using the model.

Reply:

This figure has now been moved to Section 4 where we report the simulations of ridge formation in the Gulf of Riga (It is now Fig. 5). The purpose of the figure was to display what compression ridges look like in the simulations and what they look like in reality.  We have now added a discussion on differences and similarities in the observed and simulated images.

Modified text:

"An important characteristic of sea ice break up is the formation of compression ridges. In order to demonstrate how compression ridges form in HiDEM, a square shaped sea-ice sheet of size 10 km X 10 km was modelled using DEs of 1m diameter and subjected to uniaxial compression. Figure 5A shows the outcome of this exercise. Fig. 5B shows an aerial photograph of ice ridges in the Gulf of Bothnia in March 2011 for comparison. The dynamics process of ridge formation become rather evident in these two images. Compression, induced by a strong wind, breaks up the ice in floes. Along the floe boundaries the ice fractures in compressive shear zones and ice rubble builds up to form ridges. In the simulated image, the floes still remain largely at their original position in relation to each other, while in Fig. 5B they have moved enough to form patches of open water between them."
* * *
Reviewer:
(5) Has Fig.3B been cited in the context? Is it the initial conditions for the models? Where is the straight east-west feature and why it is there?

Reply:
This figure (now Fig. 2B) was added to the manuscript to display the spatial resolution and as a demonstration that the lattice directions are not visible in the large scale fracture patterns. It is now discussed in Section 2.3. It is from the end of a simulation, not initial conditions, as now explained in the figure caption. The location of the boundary between landfast and drift ice ('the east-west feature' was a bad formulation) is now indicated in the figure.

Modified text:
"The triangular lattice structure introduces a weak anisotropy in the material stiffness and limits the crack propagation directions to a few preferred ones on the scale of a DE. The triangular lattice has three possible crack propagation directions with a 160 degrees angle between them. These angles are however not visible in the larger scale fracture patterns in e.g. Fig. 2B, which means, on a large scale the model behave predominantly isotropic, as it should."

Modified Figure caption:
"(A) The two simulation domains in Kvarken and the Gulf of Riga indicated by rectangles. (B) All DEs displayed in a 10km X 7km area in the south-western corner of the Kvarken simulation domain at a late stage of the 8/3/18 simulation when the ice is broken up. The straight boundary, from east to west, between drift and, initially stronger, landfast ice is indicated in the figure."
* * *
Reviewer:
(6) Line 140-143. Can we have some discussions on the comparisions among these Figures?

Reply:
The discussion on the results and the comparison between observed and simulated images have now been considerably extended. This discussion is now Sections 3.1 "23 of March 2018", 3.2 "8 of March 2018" , and 3.3 "Floe size distributions". We have also added new panels to the figures to better indicate details of the fracture patterns discussed in the the text.
* * *
Reviewer:
(7) What is the purpose of Figure 10? More dicusssions on the figure is necessary. It is very strange to put a figure at the end of a section without any explanations, but it happened several times in the manuscript.

Reply:
Figure 10 has now been added to a new figure (Fig.7) with several panels, including observational data on ridge formation in the Gulf of Riga. The simulation data for the location of ridges is compared to observations in Fig. 7C. Fig. 7B, shows the results of the calculation of the fractal dimension of the simulated ridge patterns, and 7D shows the distribution of wind directions and strength to demonstrate that it is predominantly south-western winds that form ridges, as assumed in the simulations. The results are now discussed in the text.

Modified text:
"To further investigate ridge formation in the Bay of Riga we identify locations of compression ridges in the simulations as places where elements are pressed below the sea surface to form ridge keels. Ridges are displayed as blue dots together with the bathymetry in Fig. 7A. It is evident from this figure that when long ridges are formed in a single event, like in our simulations, the structure

of the ridges is strongly influenced by the shape of the coast line and the bathymetry in shallow waters where ridge keels begin to get grounded. It is therefore reasonable to expect that ridge patterns form fractals, just like coastlines and many structures formed by dynamics sea ice (\cite{fractal}) do. A simple box counting algorithm, $N(L/l)\propto L/l^D$ can reveal the fractal dimension D. Here $L/l$ is the linear number of boxes the domain is divided into, and N is the number of boxes containing DEs identified as ridge keels. Fig. 7B shows the result of this exercise, which indicate that $D\approx 1.12$, which is a fairly low dimension. D=1 would mean that ridges form non-fractal linear structures. It is reasonable to expect that if ridge fields were formed over longer periods and by different wind directions they could eventually cover entire areas, and their dimension would then become D=2.  The fractal dimension D=1.12 is a rather typical value for reasonably straight coastlines like in the Gulf of Riga.

Fig. 7C shows locations of ice ridges observed from ice charts during 2000-2016. The ridge locations follow reasonably well the general ridge pattern of the simulations indicated by the reddish area in Fig. 7C. Fig. 7D shows the wind statistics (i.e. a wind rose) from the ERA5 dataset (location  58,00 and 23.75) for the same time (December 15th until May 1st in 2000-2016). This figure demonstrate that ridges are predominantly caused by SW winds, which is the dominant direction of strong winds in the area."

Reply to Reviewer2:

We would like to thank the reviewer for useful and constructive criticism of our manuscript. A 'latexdiff' version of our updated manuscript with new figures attached at the end is available as a Supplement. Below we comment on the specific points raised:
====================

Reviewer:
My first major comment concerns the fact that the analysis remains mostly qualitative. This is fine, but I think there could be a bit more discussion about the results and the way they are interpreted by the authors. For instance, Figures 4 and 6 show the motion and the compressive strains. They look interesting, but they are not commented on in the text. I think the manuscript would strongly benefit from making more explicit in the text how these figures should be interpreted. I would also recommend giving a bit more context to the comparison with observations. This could just be introducing what are the objectives set to the model results that the authors would like to demonstrate in section 3. Another suggestion could be to add a short discussion on how such high-resolution models could be evaluated, in a way like what is done for the initial conditions of the model in section 4. For instance, would it make sense to investigate the distribution of deformations? What type of observations could be used? Would there be a strong need for new observation products?

Reply:
We have now significantly expanded the text on the interpretation of the results. The Kvarken results are extensively discussed in sections 3.1. and 3.2.  We have also added some new, albeit low-resolution, observations of ridge formation in the Gulf of Riga (This is also why we have added authors: Rivo and Ilja provided these data). The Gulf of Riga results are now more extensively discussed in section 4. We have also rearranged the text a bit to improve the structure of the text that reports on the results and that which explains the model. We have now explicitly listed our objectives with the model results in the beginning of section 2.3 that describe the details of our sea ice simulations. We have also added to the Discussion, section 5, suggestions of what observational data would be useful for further evaluation of HiDEM. We also evaluate the model in Section 5 by discussing strengths and weaknesses of it, as suggested by the reviewer.
* * *
Reviewer:
My second major comment is related to the introduction of the model. This is a key part of the paper as this is the main novelty presented in my opinion, and while the first part of section 2.1 is very clear and well-written, I find the part from lines 90 to 117 more confusing. I would recommend rewriting this part section with the same philosophy that was used to write its first part: what does change in HiDEM compared to other DEM and large-scale continuum models? I think some information is missing to fully understand the results presented in section 3. I give more details on what I think should be clarified in the specific comments:

Reply:
We have now added a whole new Section 2.2. of the HPC-implementation of the code, and how it is run on high-end supercomputers. This is what really sets HiDEM apart from other DEM models. Cited papers (West et al. (2022), Damsgaard (2021), Manucharyan and Montemuro (2022)) typically report DEM results for sea ice with ~10,000 elements, while for the results reported here we use ~ 100 milllion elements.  What sets DEM models apart from continuum models is discussed

in the Introduction. We have now also considerably rewriten and expanded Section 2.3. to better explain how we model sea ice, including strengths and weaknesses of our model.
* * *
Reviewer:
L29/30: "Much more elaborate…"  This sentence is a bit unclear; do you mean these models do resolve sea ice deformations well?

Reply:
This was indeed a bit unclear. We only meant that these models are more advanced and also more accurate than the, rather crude, early models, but still have weaknesses, as all models obviously do. The sentence has been changed

Modified text:
"More advanced and more accurate continuum models are e.g. the ..."
* * *
Reviewer:
L41/42: A similar comment, these sentences are a bit unclear and seem to just be here to list some other models in a way that sounds a little bit too "casual" for a scientific journal. I would recommend just developing a bit more to justify the mention of each model.

Reply:
We have now added some text to better explain what the cited papers contain.

Modified text:
"A similar approach was later adopted by West et al. (2022) who simulated ice dynamics in the Nares strait, and by Damsgaard (2021, 2018) investigating pressure ridging. Also a recent investigation by Manucharyan and Montemuro (2022), introducing complex discrete elements with time-evolving shapes, relied on a similar approach."
* * *
Reviewer:
L68: I know this sounds pretty obvious to anyone who has worked with solid mechanics, but I would recommend giving the meaning of epsilon and sigma.

Reply:
They are now defined in the text.

Modified text:
"Here, $\sigma$ is stress and $\dot{\varepsilon}$ is strain rate."
* * *
 Reviewer:
L91: This comment is linked to my second major comment: I find the description of how these beams work a bit short and unclear, which makes it sometimes difficult to understand some of the results further in the manuscript. For instance, I still have some questions after reading this paragraph: How are these beams initialized? Do you need to have all elements connected by pair initially? Brittle elasticity is mentioned (L69), would it be possible to briefly describe how the elasticity of the beam evolves as a function of deformation? What happens after a beam is broken? Is there some sort of healing?

Reply:

All the above suggestions are now included in the section 2.3 'Sea ice simulations'. The explicit points are now explained as follows:

Modified text:
"We use close-packed spherical DEs, all of similar size, 8 meters in diameter, and connected by Euler-Bernoulli beams. A beam connect two center points of a DE. Each DE, and thereby also the endpoints of a beam, have 6 degrees of freedom: three translational, and three rotational. Beams connect all, or a fraction of randomly selected, nearest neighbors. The matrix K in Eq. (4) contains the stiffness elements (or spring constants) that relates forces and torques to beam deformation. The stiffness matrix of a single beam, and other details, are given in Åström et al. (2013). All relations between forces and deformations are linear up to a beam breaking point, which is determined by the beam deformations, either as an elastic energy criteria or as a maximum stress/strain criteria. Once a beam is broken it vanishes. I.e. the connection between the DE's is irreversibly broken, and the DE's can freely move apart but will continue to interact if they are pressed against each other. DEM parameters are listed in Table 1, and the element interactions are sketched in Fig. 1."
* * *
Reviewer:
L92: I would recommend explaining a bit more about why this anisotropy is introduced and detailing why the authors suggest that their results do not show any sort of anisotropy at a large scale. For instance, what would be expected if this anisotropy did have an effect?

Reply:
The anisotropy is a consequence of the dense-packed configuration of equal size spheres. For a single-layer sheet of spheres this will result in a triangular lattice of elements. The triangular lattice has three possible crack propagation directions at the scale of a single element. An isotropic model can be used with HiDEM, like e.g. a random packing. Such a model has the advantage of being isotropic even on the element scale, but have the disadvantage of being less dense than the close packed triangular lattice. Fig. 2B display fractured ice. If the anisotropy would strongly influence the results, the triangular lattice directions would dominate the large scale fracture pattern in this figure, which they are not. We have now tried to explain this a bit better in the text at the end of section 2.3.

Modified text:
"The typical winter sea-ice thickness in the Kvarken region is of the order of one meter, or less. It means an accurate ice thickness can only be described explicitly if the diameter of the spherical elements is no more than one meter. This would increase computational requirements immensely compared to the 8-meter spheres we use for the large scale simulations below. The number of elements would have to be increased by a factor $8^2$ to simulate the same domain. Instead, we use a single layer of DEs in a close-packed configuration forming a triangular lattice of 8 meter spheres."

"The triangular lattice structure introduces a weak anisotropy in the material stiffness and limits the crack propagation directions to a few preferred ones on the scale of a DE. The triangular lattice has three possible crack propagation directions with a 60 degrees angle between them. These angles are however not visible in the larger scale fracture patterns in e.g. Fig. 2B, which means, on a large scale the model behave predominantly isotropic, as it should."
* * *
Reviewer:
L108: This Rss ratio is described as a "governing model parameter." What do the authors mean by this expression? The authors seem to suggest it is linked to sea ice thickness and ridge formation. Would it be possible to make more explicit the physical meaning of Rss, and how the choices of

parameter values could affect the model results? Why should Rss be of order one? If Rss is not linked to ridge formation, I would strongly recommend explaining how ridge formation takes place in the model, given the emphasis on ridges in section 3. For instance, is rafting allowed? (My understanding is that it is not allowed.)

Reply:
We have now reformulated the text regarding this. Rss stands for Ratio between Stress and Strength. Fracture happens when stress on the ice is equal to or above its strength. Stress builds upp slowly until it reaches the fracture threshold, and then stress is relaxed. Therefore Rss should be close to unity when we want to simulate ice fragmentation. This is the only purpose of the Rss paramater. The reason we use it is that we cannot simulate ice with the resolution we would need to model the ice thickness that appear naturally in the Baltic sea (1m or less). This would demand element of size 1m or smaller. For a 100km X 100km simulation this would demand 10 billion particles. This could probably be done as a single large run, but would not be feasible for repeated runs. Therefore the ice is modelled as thicker (i.e. with larger elements), and therefore the applied stress on the ice must be set to match the stonger ice.

This relates to ridge formation only because ridge formation becomes more difficult with larger elements.

Ridge formation in the model is now discussed at the beginning of Section 4.

Modified text:
"An important characteristic of sea ice compression is the formation of pressure ridges. In order to demonstrate how pressure ridges form in HiDEM, a square shaped sea-ice sheet of size 10 km × 10 km was modelled using DEs of 1m diameter and subjected to uniaxial compression. Figure (5A) shows the outcome of this exercise. Fig. (5B) shows an aerial photograph of ice ridges in the Gulf of Bothnia in March 2011 for comparison. The dynamics process of ridge formation become rather evident in these two images. Compression, induced by a strong wind, breaks up the ice in floes. Along the floe boundaries the ice fractures in compressive shear zones and ice rubble builds up to form ridges. In the simulated image, the floes still remain largely at their original position in relation to each other, while in Fig. (5B) they have moved enough to form patches of open water between them."

Reply:
Rafting appear in the model. We have now added an animation as Supplementary material to the manuscript to better demonstrate the model simulation outcome, including rafting of ice floes.
* * *
Reviewer:
L163: "This originates from…" I would suggest adding "most likely", given the authors do not demonstrate it.

Reply:
We used this formulation because we practically know that this is the case. We have nevertheless modified the text as suggested.
* * *
 Reviewer:
Figure 8C,D: What are the dimensions of the x-axis, metres or metres squared? This is a bit unclear.

Reply:

The dimension is numbers of elements in a floe. One element has an area of 50 square meters so the dimension is 50 m2. We have no clarfied this better in the text.

Modified text:
"The scale on the x-axis is number of elements, DEMs, in a floe (1 DEM \approx 50 m^2 )."

---

## Author Response (AR2)

Dear editor

I have now edited the manuscript according to your suggestions. In order to increase the sizes of Figures 3 and 4 I had to split them into several figures. I have also reformulated 'Discussion' and 'Conclusions' sections as recommended.

Kind regards, Jan Åström